# Impact of Acute Lymphoblastic Leukaemia Treatment on the Nutritional Status of Paediatric Patients: A Systematic Review

**DOI:** 10.3390/nu16234119

**Published:** 2024-11-28

**Authors:** Rosaura Picáns-Leis, Fátima Nieto, Anabel Romero-Agrelo, Isabel Izquierdo-López, Lucía Rivas-Rodríguez, Rocío Vázquez-Cobela, Rosaura Leis

**Affiliations:** 1Neonatology Department, University Clinical Hospital of Santiago de Compostela, Santiago de Compostela University, 15706 Santiago de Compostela, Spain; rosaurapicansleis@gmail.com; 2Health Research Institute of Santiago de Compostela (IDIS), 15706 Santiago de Compostela, Spain; fatima.nieto.pombo@sergas.es (F.N.); aromeroagrelo@gmail.com (A.R.-A.); isabel.izquierdo.lopez@gmail.com (I.I.-L.); luciarivas@hotmail.com (L.R.-R.); cobela.rocio@gmail.com (R.V.-C.); 3Unit of Pediatrics Gastroenterology, Hepatology and Nutrition, Pediatrics Department, University Clinical Hospital of Santiago de Compostela, Santiago de Compostela University, 15706 Santiago de Compostela, Spain; 4Consorcio CIBER, M.P. Fisiopatología de la Obesidad y Nutrición (CIBEROBN), Instituto de Salud Carlos III (ISCIII), 28029 Madrid, Spain

**Keywords:** systematic review, leukaemia, children, treatment, nutritional status, nutrition assessment

## Abstract

Introduction: Acute lymphoblastic leukaemia (ALL) is the most prevalent cancer in childhood and is one of the leading causes of death annually. Antineoplastic treatments are associated with a high risk of malnutrition, which is important for continuous growth and development. Objective: This systematic review aimed to evaluate the effect of these treatments on the nutritional status of paediatric patients with ALL. Data were collected from the PubMed, Scopus, and Web of Science databases following the Preferred Reporting Items for Systematic Review and Meta-Analysis (PRISMA) method. All the studies in which nutritional status was assessed in human beings diagnosed with ALL up to 18 years and published in English or Spanish between January 1900 and December 2023 were included. Results: A total of 18 articles and 1692 participants were analysed in this review. Article quality was evaluated using the ROBINS-I tool. This systematic review has been registered on PROSPERO (CRD42024505111). ALL treatment has a negative impact on the nutritional status of these patients and played an important role in their growth and global development. ALL treatments can modify hunger perception and the palatability of food. Conclusions: Nutritional status is important for patient prognosis and survival. Nutritional evaluation, a specific nutritional assessment, and follow-up should be recommended for these patients to decrease the nutritional impact on global health. More homogeneous studies must be conducted to reach robust conclusions regarding the best type of nutritional intervention for these patients.

## 1. Introduction

Acute lymphoblastic leukaemia (ALL) is the most prevalent type of cancer and the leading cause of cancer-related deaths in children [1], although its 5-year-after-diagnosis survival has increased to 80% in recent years [2]. The highest incidence of ALL diagnoses was found between the ages of 2 and 5 years.

ALL can be classified according to the risk at diagnosis, depending this classification on the criteria used for stratification. In 1993, the Paediatrics Oncology Group (POG) and Children’s Cancer Group (CCG) established a criterion to set the risk in two risk groups: standard risk (SR) for patients between 1 and 10 years and with a white blood cell (WBC) count <50 × 10^9^ cells/L, and high risk (HR) for the other ALL patients, including T-ALL. Other groups have developed other standards for risk-stratified treatment approaches, including the St. Jude Consortium, the Dana–Farber Cancer Institute (DFCI) ALL Consortium, and the Berlin–Frankfurt–Münster (BFM) Group [3].

ALL treatment is adapted to the patient’s specific risks, since adherence to the treatment schedule (indirectly) also has an impact on nutritional risk [4].

In general, the treatment for ALL consists of 3 basic phases. The first one is the Induction, whose objective is the reduction of the tumour size as much as possible and takes around four weeks. The following phase is the postinduction Consolidation which is based on the elimination of remaining potentially leukemic cells with a variable duration of up to eight months. The last one is the Maintenance, indicated to prevent relapse of the disease, which lasts for around 2–3 years post-treatment, depending on the patient characteristics. All these phases include the use of antineoplastic medication and corticosteroids depending on the risk and the protocol applied [3], as well as cranial irradiation in some cases. Prophylactic cranial irradiation (CI) has been a standard treatment in paediatric patients with ALL at high risk for central nervous system (CNS) relapse [5], although there is a current great consensus that most patients do not need it [6]. The disease itself has a negative effect on nutritional status as cancer cells involve alterations to some metabolic pathways, such as glycolysis, glutamonolysis, and lipogenesis [7]. In addition, antineoplastic treatments greatly affect the general condition of these patients, their susceptibility to infections, and their nutritional status, which is particularly important in this vulnerable population owing to their state of growth and development [8].

After diagnosis, many factors play an important role in nutritional impairment in these children. Antineoplastic treatments can influence changes in nutritional status, with significant differences between the different stages of treatment, with the early stages being associated with the highest risk [3,9]. Specifically, corticosteroids and CI used in protocols against ALL are well known to be related to the regulation of energy intake and impair signalling reception from hormones that regulate hunger and appetite. This results in several effects on body composition, leading to short- and long-term impacts on the nutritional status of survivors [10]. Also, as a consequence of the treatment, the motor function of ALL patients becomes impaired, extending this condition to adolescence and adult age [11], making physical activity difficult.

Some studies have demonstrated that ALL therapies in childhood are linked to a greater risk of chronic metabolic diseases, probably increased, among other reasons, as a consequence of weight gain and obesity. Also, it was observed that these conditions are present fundamentally in patients who are overweight/obese at diagnosis and who retain a high BMI z-score during the treatment and until the beginning of survival [12]. We must take into account that paediatric age is a key moment of intervention for the acquisition of a healthy lifestyle and nutritional habits that can play an important role in short-, medium-, and long-term health, as well as in the prognosis of different illnesses [13].

When performing a nutritional assessment, anthropometric parameters, intake and metabolic biomarkers, diet, physical activity, and lifestyle surveys should be carried out [14,15,16]. The capacity of these factors on metabolic programming from very early ages to adulthood is well-known [15,16]. Thus, nutritional intervention and the establishment of a healthy lifestyle should be priority objectives and must be considered a main part of the multidisciplinary treatment of high-risk patients. Additionally, undernourishment in the paediatric age in industrialised countries is now associated with the disease. For this reason, is essential to highlight the necessity of adequate adherence to clinical practice guidelines for nutrition and physical activity in children with ALL.

Therefore, this systematic review aimed to identify the effects and influences of antineoplastic treatments on the nutritional status of children diagnosed with ALL.

## 2. Materials and Methods

This review was registered in the International Prospective Register of Systematic Review (PROSPERO) with the ID code CRD42024505111. It was conducted according to the Preferred Reporting Items for Systematic Review and Meta-Analysis (PRISMA) checklist [17].

The review question was “Does acute lymphoblastic leukaemia treatment influence the nutritional status of paediatric patients?”. The Population, Intervention, Comparison, and Outcome (PICO) model [18] was used (Table 1).

### 2.1. Search Strategy

A systematic search was conducted to identify relevant publications on the effects of ALL treatment on the nutritional status of children. This study was conducted by consulting the approved scientific databases PubMed, Scopus, and Web of Science.

The search formulas used in the PubMed, Scopus, and Web of Science databases were created using keywords and Boolean markers AND, OR, and NOT. The keywords were selected using Medical Subject Headings (MeSH) terms in PubMed and tools, such as asterisks, to use general terms related to search concepts from the Scopus and Web of Science databases. Thus, the search formulas presented in Table 2 were developed.

### 2.2. Inclusion and Exclusion Criteria

This review included publications in (1) English or Spanish (2) between 1 January 1900 and 31 December 2023 which (3) based their study on human beings with a (4) diagnosis of ALL and (5) who have been treated against ALL, (6) aged less than or equal to 18 years, and (7) presented two or more measures for the assessment of nutritional status. The studies were screened based on the type of scientific publication, selecting only clinical trials.

Articles that did not meet the inclusion criteria were excluded. Studies conducted on survivors or those that used any type of supplementation or parenteral nutrition were excluded.

### 2.3. Selection of Studies

Two of the authors (R.P.-L. and F.N.) selected the articles included in this review by searching the PubMed, Scopus, and Web of Science databases. This selection was reviewed by the other authors (A.R.-A., I.I.-L., L.R.-R. and R.V.-C.). In cases where a consensus was not reached, R.L. acted as a referee. First, duplicates were excluded using the web application Rayyan [19]. Two authors independently reviewed the titles and abstracts to identify potentially relevant articles according to the inclusion and exclusion (all articles not meeting the inclusion requirements) criteria. Independently, again manually, information was extracted from the selected studies after reading the full text to assess their eligibility following the same criteria.

### 2.4. Data Extraction

A data extraction form was used to assess the quality and synthesis of evidence. The extracted information included the following data: author, sample (n and years), population, study design, objective, outcomes, treatment, comparison, results, and conclusion, as well as any other data considered relevant to our systematic review. This information was afterwards synthesised in Table 3.

### 2.5. Quality Assessment and Bias Analysis

The assessment of bias risk was carried out by two authors independently using ROBINS-I tool [20], a Cochrane Library tool to evaluate the risk of bias in non-randomised studies of interventions. A summary and a graph were created using the Risk of Bias Visualization tool Robvis [21] (Cochrane Collaboration), which allows for bias stratification in a few domains to certify whether the studies included in the review had high, moderate, low, or no information risk.

### 2.6. Data Synthesis

A formal narrative synthesis was used to provide a synthesis of the results from the included studies, structured according to their main objectives, anthropometric variables and biomarker analysed, as well as healthy lifestyles.

**Table 3 nutrients-16-04119-t003:** Main characteristics of the articles included in the systematic review.

Reference	Sample (n, Age)	Population	Type of Study	Objective	Outcomes	Treatment	Comparation	Results	Conclusion
Tamminga et. al (1992) [22]	n1 = 7 n2 = 13 n3 = 13 n4 = 8 nT (complete remission): 41 (2.4–13.4 years ^1^) 5.9 ^2^	The Netherlands	Longitudinal	To explore potential variations in growth impairment among patients undergoing chemotherapy, both with and without cranial irradiation.	Weight, height, weight for height, upper arm circumference, sitting height, arm span, and head circumference	Group 1: HR protocol Group 2: SR protocol + CI Group 3: SR protocol + CI substituted with regimens of high-dose methotrexate and more intensive intrathecal chemotherapy. Group 4: SR protocol + CI HR protocol: consolidation with cytarabine and cyclophosphamide SR protocol DCSG	Anthropometric variables investigated at diagnosis and every 3 months until 2 years after diagnosis (G1-3) or until 2 years after completion of therapy (G4).	- Retardation in height of 0.4–0.6 SD during therapy in G1–G3. Catch-up of 0.5 SD in G4. - Significant difference in sitting height only in G1 and G4.00 - Arm span growth was delayed in G1–G3, showing a pattern similar to height, although the delay appeared greater compared with height or sitting height. G4 demonstrated catch-up growth. - Notable weight gain exceeding normal levels was observed in G2–G3. - Head circumference remained unaffected. - Excessive weight gain was observed in all groups.	Average height delay occurred during the 2 years of treatment, with the most significant retardation observed in the first six months after diagnosis. Height catch-up was anticipated at the end of therapy, aligning with catch-up growth in sitting height and arm span. Cytostatics, corticosteroids, and cranial irradiation (CI) might be linked to this reduction. The weight gain could be associated with supplementary tube feeding in some patients, the use of corticosteroids, and psychological factors.
Halton et al. (1998) [23]	Study 1 A: n: 55 B: n: 42 C: n: 19 nT: 116 Study 2 n: 16 Study 3 n = 19 nT = 116	Canada	Longitudinal	To analyse the effects of the disease and its treatment on growth and body composition in children with ALL from population-based referral regions in Canada.	Height, weight, faecal fat, D-xylose, and lactose absorption and dietary intake by questionnaire. Serum albumin. Fat-free mass and bone density (DEXA)	DFCI protocol Study 1: A: HR. 87-001 protocol + CI B: SR. 87-001 protocol C: SR. 91-001 protocol (8 males with CI and 11 females without CI) Study 2: Protocol 87-001 Study 3 Protocol 91-0001	Study 1: altered growth in children treated Height and weight: Diagnosis, every 12 months during treatment, and 12 months following the end of therapy. Study 2: impact of chemotherapeutic agents on intestinal functional integrity Nutrient intake, faecal fat, and D-xylose absorption and lactose absorption Measurements at diagnosis and every 6 months during therapy. Study 3: nutritional status and body composition during treatment. Anthropometric measurements. Serum albumin and total protein. Alterations in body composition, including changes in lean mass, fat mass, and bone mineral content.	Study I - There was a decrease in mean height during the first year in all groups. A further decline in height occurred in the second year only in the groups that received CI. Catch-up growth was observed in all groups after therapy completion, although children who received CI showed a smaller increase in mean height. - Dietary intake remained consistent throughout the two-year therapy period and was not associated with the decrease in growth. - Disproportionate increase in weight compared with height during the second and third year, showing tendency towards obesity. Study II - Decrease of height velocity in 73% children at 12 months and 29% at 24 months. - A decrease in weight velocity was observed in 40% of cases at 12 months and 14% at 24 months. - Nutrient intake was above 2/3 of the RNI for energy in 70% of children, and exceeded 100% of the RNI for protein in all but one child. - D-xylose absorption was abnormal in 2 children at 6 months of therapy, but normalised by 12 months and remained normal thereafter. Intestinal lactose absorption was impaired in 4 out of 12 children. Study III - Decreased height observed during treatment. - Decreased albumin levels were observed in 32% of cases at diagnosis and 78% during intensive consolidation therapy. - A 5% reduction in the ratio of lean mass to total body weight was observed by 6 months of therapy. - Body fat increased by 6%, from 22% at diagnosis to 28% at the completion of therapy. - Therapy was compromised in the majority (5/6) of high-risk (HR) patients during the consolidation phase, while standard-risk (SR) patients tolerated it better.	The majority of children treated for ALL experienced significant changes in nutritional status, including decreased growth, alterations in body composition (decreased lean mass and increased fat mass), and lower serum protein levels during intensive therapy. It may have been caused as a response to growth hormone and the use of corticosteroids. Reduced physical activity during treatment, secondary to musculoskeletal morbidity, which reduced energy expenditure and led to weight gain. Malabsorption implicated in contributing to growth impairment as cytotoxic agents cause changes in the intestinal mucosa and malabsorption of D-xylose and lactose. Decreased albumin levels probably due to depletion of visceral proteins at diagnosis and to the combination of chemotherapy with L-asparaginase during treatment.
Sgarbieri et al. (2006) [24]	n = 45 16 girls and 29 boys (1–11 years ^1^) 5 ± 2.6 ^2^	Brazil	Longitudinal	To assess the nutritional status, growth patterns, and serum zinc and copper concentrations at the time of diagnosis and throughout the treatment of children with ALL.	Weight, height, W/H, and main daily intakes (translated to calories, protein, zinc, and copper)	Chemotherapy GBTLI-ALL protocol CI HR patients during reinduction	Group HR Group SR Anthropometric measurements, daily intake, and serum trace elements were assessed four times during the study period: at diagnosis, and during induction, reinduction, and maintenance phases. Mean follow-up was 18 months	DIAGNOSIS - 5 patients were malnourished according to anthropometric indices at diagnosis. - A decrease in calorie intake was observed in 76% of the children, while only 9% had protein intakes below the recommended daily allowance (RDA). - Zinc and copper: 24% and 2% of the children had intakes <DRI. TREATMENT - Induction and re-induction: Increase in weight - Maintenance: A decrease in weight was particularly noticeable among the group of children with standard risk (SR). - A decrease in mean growth rate during treatment was observed, particularly among children with high risk (HR) who received prophylactic intracranial (IC) therapy. - Dietary intake showed an increase in calories, protein, zinc, and copper during the induction and reinduction phases, followed by a decrease in intake during the maintenance phase. - Serum copper levels decreased during the induction phase and remained stable throughout the rest of the treatment.	The increased energy intake and weight gain may be attributed to the high-dose steroids administered over several weeks. Additionally, water retention could be linked to the increase in fat. Nutritional status changes were observed, including reduced growth, weight fluctuations, and alterations in copper and zinc levels.
Higashiyama et al. (2014) [25]	n = 23 (9 SR, 14 HR) 19 boys and 4 girls (1.3–14.6 years ^1^) 4.5 ^2^	Japan	Longitudinal	To assess changes in anthropometric measurements and serum albumin levels during chemotherapy for ALL.	Weight, height, BMI z-score, Waterlow score, serum albumin, and CRP levels.	Chemotherapy Treatment used: JCLSG	Measurements were taken at the beginning and end of each chemotherapy phase.	- At diagnosis, 2 patients (8.7%) were categorised as underweight and 5 (21.7%) as overweight according to the BMI z-score. When assessed using the Waterlow score, 5 patients were classified as underweight and 3 (13%) as overweight. - A significant decrease in the absolute values of the BMI z-score and waist-to-height ratio (W/H) was observed during the sanctuary and re-consolidation phases, particularly in high-risk (HR) patients. - A significant decrease in albumin levels was observed during the induction and re-induction chemotherapy phases. - Hypoalbuminemia was present in 13 patients (56.5%) during the induction phase and in 8 patients during the re-induction phase. At diagnosis, 2 patients exhibited hypoalbuminemia. - A significant decrease in daily energy intake was reported during chemotherapy. - There was no correlation between albumin and CRP levels.	Hypoalbuminemia may have been induced by L-asparaginase (used during induction and re-induction), as it acted as an inhibitor of protein synthesis. A decline in nutritional status was observed across different phases of chemotherapy, as indicated by the changes in BMI z-score, Waterlow score, and albumin levels.
González et al. (2004) [26]	n = 49 23 girls and 26 boys (1–15 years ^1^) 6 years ^2^	Cuba	Longitudinal	To assess the nutritional status of children receiving treatment for ALL.	Height, weight, BMI, and skin folds at the suprailiac, subscapular, and tricipital sites.	Chemotherapy Protocol I-LLA-90 of GLATHEM	Time points for evaluation were at diagnosis, after the intensive phase, and at the end of treatment.	- 39% of the patients increased in weight percentile after the intensive phase of treatment - No differences in skin-fold measures in more than 1/3 of the children. - Similar results at the end of therapy - Tendency to increase weight without radiotherapy, more obvious in boys.	No significant effect of chemotherapy on nutritional status was observed in this Cuban paediatric population.
Morrison et al. (2011) [27]	n = 11 39–171 months ^1^	USA	Longitudinal	To evaluate the correlation between creatinine and lean body mass in patients with ALL.	Weight, height, BMI, biochemical markers, fat-free mass, and fat mass.	Chemotherapy Protocol DFCI protocol 20-01 Treatment for 25 months	Measurements were taken at diagnosis, 6 months, 12 months, 18 months, 24 months, 30 months, and 36 months.	- Anthropometry: At diagnosis, the overall mean BMI was within the normal range. However, 6 months after diagnosis, during the most intensive therapy period, the median BMI Z-score dropped to −0.6 standard deviations. A progressive recovery was observed thereafter. - Body composition: At diagnosis, the mean fat mass and body mass were within normal limits. Fat mass Z-scores ranged from −1 to +1, while body mass Z-scores showed greater variability, ranging from approximately −2 to +2. During the first 6 months of treatment, there was an overall increase of +0.63 in the fat mass Z-score. However, a decrease of −0.99 was observed in the fat mass Z-score thereafter. The mean fat mass and lean body mass (LBM) Z-scores remained within normal ranges. - Biochemistry: A strong correlation was found between urine creatinine and lean body mass (LBM) (r = 0.79, *p* < 0.001). Linear regression analysis indicated a slope of 20.9 ± 1.6 kg of LBM per gram of urine creatinine per day. However, the correlation between serum creatinine (mML1) and LBM over the entire 36-month period was weaker (r = 0.52, *p* < 0.001).	BMI decreased by −0.6 SD during the most intensive therapy but was followed by progressive recovery. In the first 6 months of treatment, there was a weight gain, with LBM loss and fat mass increasing. A statistically significant correlation was found between 24 h urine creatinine levels and lean body mass in ALL patients.
Gokhale et al. (2007) [28]	n ALL: 105 n C: 108 nt = 213 (3–17 years ^1^) 5.25 ± 3.45 ^2^	India	Longitudinal	To determine serum levels of cholesterol, retinol, zinc, albumin, and haemoglobin in children who had completed treatment for ALL.	Weight, height, and biochemical parameters (albumin, cholesterol T, retinol, zinc, and haemoglobin)	Chemotherapy Protocol NCI Treatment: MCP-841 CI during Induction	Comparison of parameters by stage of inclusion and by type of treatment, as well as the patient and control group.	- Serum albumin levels at 6 months (M6) were significantly higher than those of post-therapy patients (t = 2.31, *p* = 0.05). - No changes were observed in the other biochemical parameters between patients at M6 and those post-therapy. - More than 75% of patients and controls had deficient serum retinol levels: 67/88 patients and 41/75 controls (difference not significant)	There were no long-lasting effects of treatment on serum albumin, total cholesterol, retinol, zinc, or haemoglobin levels. The majority of subjects with low serum retinol levels indicated depleted liver reserves. Low serum retinol levels in at least 75% of both patients and controls likely reflected poor dietary intake. Additionally, the higher percentage of patients with low retinol levels may have been attributed to urinary losses of retinol during infection and the use of immunosuppressive drugs.
Delbecque et al. (1997) [29]	nALL: 15 nC: 15 nT: 30 1.3–14.6 years ^1^	France	Longitudinal	To assess the impact of ALL treatment on the nutritional status of patients.	Skin folds (biceps, triceps, subscapular, and suprailiac) and bioimpedance	Chemotherapy Protocol 58881 EORTC	Measurements at diagnosis, 22 days, 36 days, and 71 days.	- The mean values for the body size and composition of the patients were similar to those of the control subjects on the day of diagnosis. Three patients were categorised in the malnourished range, with two patients below 85% (83.9% and 82.6%) and one patient below 80% (63.6%) of the French standards for weight-for-height. These results remained consistent on days 22, 36, and 71. - There was no difference in fat-free mass (FFM), whether estimated using skinfold thicknesses or bioelectrical impedance analysis (BIA). FFM obtained through anthropometry showed a strong correlation with FFM measured by BIA. - Reduced energy intake was observed in patients on day 1 compared to control subjects, with this difference persisting until day 22, but disappearing by days 36 and 71. On day 1, 9 out of the 15 patients were consuming less than 80% of the French recommended daily allowance (RDA). By day 71, only 5 patients had intakes below 80% of the French RDA, with 4 of them exceeding 100%. The low energy intake involved both carbohydrates (on days 1 and 22) and fats (on days 1, 22, and 36). However, all patients had protein intakes above the French RDA, although these were significantly lower than those of the control subjects on day 1. Significant variations in energy and carbohydrate intake were noted from day 1 to day 71, while no significant changes were observed in fat and protein intakes. - The respiratory quotient (RQ) of the ALL patients on day 1 was significantly lower than that of the control subjects. However, this difference between the two groups was no longer present on day 71. - Cytokines were undetectable in the serum on days 1, 36, and 71.	Reduced energy intake was observed in ALL patients compared with controls. However, dietary intake increased during treatment, particularly in terms of energy and carbohydrates, with all patients consuming protein levels above the normal range. Only 5 patients had intakes below 80% of the French RDA. Energy expenditure was lower in sick patients, stabilising by the end of treatment. No changes in body composition were noted throughout the treatment period.
Bond et al. (1992) [30]	nc = 26 nALL = 16 5–16 years ^1^	UK	Cross-sectional	To assess basal energy expenditure during treatment	Basal metabolic rate, weight, height, and weight for height	Chemotherapy MRCUK	Measurements of energy intake and basal metabolic rate were taken during the week prior to the chemotherapy session.	- The mean body size and composition values of the patients were comparable to those of the control group. - One patient with ALL was classified as stunted, with a height for age below 90% of the expected value. - The range of energy intake values in the chemotherapy group (85%) was not significantly different from that of the control group (87%).	No significant differences were observed between the control group and ALL patients in terms of basal energy expenditure or energy intake. There were no signs of inadequate nutritional status.
Ladas et al. (2019) [31]	nt = 794 (1–18 years ^1^)	USA	Longitudinal	To understand the influence of paediatric ALL treatment on patient’s dietary intake	Weight, height, and BMI. Food Frequency Questionnaire (HSFFQ) and Portions Questionnaire (YAFFQ).	Chemotherapy	They compared dietary intake at diagnosis and throughout treatment, with RDI and normative values.	- At each timepoint, dietary intake was recorded for 81% (n = 640) of participants at diagnosis, 74% (n = 580) at the end of the induction phase of treatment, and 74% (n = 558) during the continuation phase. Despite exposure to corticosteroids, caloric intake decreased throughout therapy for most age and gender groups. - Predictive models of excess intake indicated decreased odds of over-consuming calories (OR = 0.738, *p* < 0.05), but increased odds of over-consuming fat (OR = 6.971, *p* < 0.001). - When compared with NHANES data, we consistently found that more than one-third of children were consuming calories in excess of normative values. - For micronutrient selection, a small proportion of participants were either above or below the dietary reference intake (DRI) at each timepoint assessed.	The study suggests that dietary intake varies during treatment for ALL when compared with age- and gender-specific recommended and normative values. Enhancing our understanding of nutrient fluctuations and dietary quality will support future analyses exploring the relationships between dietary intake, toxicity, and survival outcomes.
Sala et al. (2005) [32]	n = 37 (SR + 14 HR)	Canada	Longitudinal	To assess the relationship between serum creatinine levels and lean body mass.	BMI, serum creatinine, urine creatinine, and lean body mass by dual-energy X-ray absorptiometry.	Chemotherapy DFCI HR: 3× cumulative dose of corticosteroid of SR	Two cohorts of patients were analysed. The study included 37 children with ALL and 20 children with primary muscular disorders (PMD), with the latter group serving as a comparison cohort.	- A strong correlation was observed between serum creatinine and lean body mass in patients with PMD (r = 0.77), as well as in patients with ALL (r = 0.83 at diagnosis, r = 0.77 during therapy, and r = 0.56 after therapy). - The correlation between serum creatinine and body size (body mass index) was much weaker, with values of r = 0.38, r = −0.09, and r = 0.29 at successive observations in the ALL cohort, and r = 0.05 in the primary muscular disorders cohort.	Further research involving a broader range of diseases is needed, as this would enable the consideration of serum creatinine as a general surrogate marker for lean body mass and, by extension, nutritional status.
Andria et al. (2020) [33]	n = 36 ALL patients 1–18 years ^1^	Indonesia	Longitudinal	To observe the occurrence of abnormal blood glucose levels in children with ALL treated with prednisone and L-asparaginase and its effect on the state of remission.	Fasting glycemia and 2 h postprandial glycemia	Chemotherapy	From the time of established diagnosis to the end of the induction phase of chemotherapy (8 weeks).	- Blood glucose tests during the induction phase revealed hyperglycemia in 6 subjects, hypoglycemia in 9 subjects, both hypoglycemia and hyperglycemia in 7 subjects, and euglycemia in 14 subjects. The onset of abnormal blood glucose levels is depicted. - 7 subjects did not achieve remission after completing the induction phase.	From 36 patients: 6 hyperglycemia,9 hypoglycaemia, 7 hypoglycaemia and hyperglycaemia and 14 euglycemia. 5/7 children who did not reach remission had abnormal BG. Abnormal BG levels were related to poor nutritional status, but did not affect disease remission.
Aisyi et al. (2019) [34]	n = 57 0–18 years ^1^	Indonesia	Analytic prospective study	To determine the incidence of hyperglycemia during the induction phase of chemotherapy in children with ALL, specifically comparing the effects of prednisone and dexamethasone (in combination with L-asparaginase) on the development of hyperglycemia.	BMI and blood glucose level	The induction phase of chemotherapy was conducted following the Indonesian Protocol for Acute Lymphoblastic Leukemia 2013	The third week of treatment and once every week up to the sixth week of the protocol.	- The median blood glucose level (BGL) in the 4th week was 91.00 (range 58–263), while the median in the 6th week was 86.00 (range 50–651).	No correlation was found between overweight patients and the incidence of hyperglycemia. The incidence of hyperglycemia was 5.2%. Dexamethasone, in combination with L-asparaginase, was associated with a higher risk of altering blood glucose levels compared with prednisone.
Santoso et al. (2010) [35]	n = 40 N Group A = 20 N Group B= 20 5–18 years ^1^	Indonesia	Cross-sectional comparative	To obtain the proportion of BMD, calcium, and vitamin D levels in children after 6 and 12 months of chemotherapy for ALL	Bone mineral density. Vitamin D3 and calcium levels, measured in group A at 6 months and in group B at 12 months	Chemotherapy maintenance phase in accordance with the National Protocol (Jakarta) for high risk ALL.	Group A: children who had been treated for 6 months chemotherapy maintenance phase. Group B: children who had been treated for 12 months of chemotherapy maintenance phase	- A total of 40 subjects enrolled in this study. - The incidences of hypocalcemia and vitamin D deficiency were 33 out of 40 and 40 out of 40, respectively. - The mean calcium ion levels, 25(OH)D3 levels, and BMID z-score values in the 6-month group were 1.1 (0.1 SD) mmol/L, 21.3 (2 SD) ng/L, and −0.7 (0.8 SD), respectively. In the 12-month group, the values were 1.1 (0.0 SD) mmol/L, 21 (2.2 SD) ng/L, and −1.7 (0.6 SD), respectively (*p* = 0.478). - Body mass index (BMI) and cumulative corticosteroid dose were correlated with low bone mineral density (BMD) values in the L1–L4 vertebrae.	Hypocalcaemia (33 of 40), vitamin D deficiency (40 of 40), have been observed during chemotherapy in patients with ALL. BMD in L_1_ and L_4_ is correlated with corticosteroid treatment and BMI. 23 of them maintained a normal bone mineral density until the end of the treatment, while 15 of them were diagnosed with osteopenia and 1 with osteoporosis.
Roy et al. (2017) [36]	nt = 52 7 ± 3.3 years ^1^	India	Longitudinal	To determine the prevalence and clinical implications of folate deficiency in children with ALL during the maintenance phase of treatment.	Folate and dietary pattern albumin.	Chemotherapy with antifolate treatment	The study investigated the prevalence and clinical implications of folate deficiency during the maintenance phase of treatment, considering the prolonged use of antifolate medications and the high prevalence of folate deficiency in the population.	- 29 out of 52 children enrolled in the study experienced folate deficiency at some point during maintenance chemotherapy. - Neutropenia (18 of 29 vs. 4 of 23; *p* = 0.002), thrombocytopenia (17 of 29 vs. 4 of 23; *p* = 0.005), febrile neutropenia (17 of 29 vs. 4 of 23; *p* = 0.005), and the need for chemotherapy dose reduction (20 of 29 vs. 7 of 21; *p* = 0.01) were more common in children with folate deficiency. - Mortality during the maintenance phase was higher among deficient children (8 out of 29 vs. 1 out of 23; *p* = 0.03), and their survival rates were significantly lower (*p* = 0.02). Multivariate analysis revealed that hypoalbuminemia (*p* = 0.02) and folate deficiency (*p* = 0.01) were linked to febrile neutropenia, while folate deficiency was also associated with maintenance phase mortality (*p* = 0.03).	Folate deficiency was linked to treatment-related complications and poor outcomes in our patients. The risks and benefits of folate supplementation for deficient children during maintenance chemotherapy should be further investigated through well-designed randomised studies in similar settings.
Pais et al. (1990) [37]	Cohort 1 n ALL = 11 n C = 11 nt = 22 (3–16 years ^1^) Cohort 2 n ALL = 26 nC = 26 nt = 52 (2–18 years ^1^)	USA	Cross-sectional	To identify differences in vitamin B6 levels in the control group and patients with ALL due to its role in the immune system and cellular control.	Weigh, height, and diet history to measure B6 intake.	Chemotherapy	The comparison was performed with a control group consisting of healthy siblings (n = 26) of cancer clinic patients, matched by age.	- Plasma PLP levels were markedly reduced in children with leukaemia compared with healthy controls. - There was no correlation between PLP levels and the absolute count of circulating leukemic cells in peripheral blood. - Plasma PLP showed a positive association with vitamin B6 intake. Serum prealbumin levels were significantly diminished in children with leukaemia compared with controls. - Indicators like albumin levels, weight loss, and percentage of ideal body weight were markers of chronic poor nutrition. - Plasma PLP levels showed no significant correlation with serum prealbumin, serum albumin, weight loss, percentage of ideal body weight, or overall nutritional status. - Additionally, no association was found between PLP levels and liver transaminase enzymes. ALT and AST levels remained entirely normal in all patients with low plasma PLP levels (<7 ng/mL) who presumably had adequate vitamin B6 intake.	The suboptimal vitamin B6 status cannot be attributed to antineoplastic therapy, as all samples were taken at the time of initial diagnosis, prior to chemotherapy. Control siblings had PLP levels similar to normal values reported elsewhere, indicating that most subjects likely did not have an undetected genetic or environmental predisposition to vitamin B6 deficiency before developing leukaemia. Low plasma PLP levels are considered more likely to result from leukaemia itself rather than being a cause. This could be explained by reduced vitamin B6 intake during the illness or impaired hepatic conversion of pyridoxine to PLP.
De Carvalho et al. (2020) [38]	n = 14 (<19 years) 7 years ^2^	Brazil	Longitudinal	To evaluate changes in nutritional status, food intake, and appetite-regulating hormones in children and adolescents with acute lymphoblastic leukaemia during the first phase of chemotherapy.	Weight, height, arm circumference, and arm skin fold. Dietary intake. Ghrelin, insulin, leptin, and serum cortisol.	Induction chemotherapy	Anthropometric measurements, 24 h dietary intake records, and appetite-regulating hormone levels were evaluated at three time points: prior to, midway through, and at the conclusion of the induction phase.	- The majority of patients (85.7%) had a normal weight at the start of treatment, with no significant changes observed over the 28-day period. - Energy and nutrient intake showed improvement from the beginning to the midpoint of treatment, as indicated by the increase in ghrelin levels (from 511.1 ± 8.3 to 519 ± 6.6 pg/mL; *p* = 0.027). - No significant changes were noted in other appetite-regulating hormone levels.	Food consumption improves during the initial phase of treatment, without significant changes in anthropometric measures of nutritional status.
Moschovi et al. (2008) [39]	n ALL = 9 n C: 9 nt: 18 (2–7 years ^1^) 4.3 ^2^	Greece	Longitudinal	To determine variations in the PYY and ghrelin levels during treatment.	BMI, serum samples, PPY, serum ghrelin	Chemotherapy treated with HOPDA 97 protocol	BMI, leukemic load, and levels of PYY and ghrelin were measured at diagnosis, following the induction-consolidation phase, and at standard intervals prior to each treatment cycle in 9 patients with ALL. These measurements were compared to those of 9 healthy, age- and sex-matched control children.	- At diagnosis, mean PYY levels were significantly elevated (*p* < 0.0001), while mean active ghrelin levels were notably reduced compared with controls (*p* < 0.001). - Following the induction–consolidation phase, PYY levels increased significantly compared with baseline (*p* = 0.033) but gradually returned to pretreatment levels by the sixth chemotherapy cycle. - Ghrelin levels, in contrast, fluctuated during treatment and stabilised at significantly higher levels (*p* = 0.024) after the eighth cycle. However, even after the eighth cycle, ghrelin levels remained lower than those observed in controls (*p* < 0.001).	PYY and ghrelin are key contributors to the development of anorexia–cachexia syndrome in paediatric patients with ALL.

Abbreviations: ^1^, age range; ^2^, mean age; SR, standard risk; HR, high risk; ALL, acute lymphoblastic leukaemia; CI, cranial radiotherapy; Cint, intensive chemotherapy; G, group; C; control; Pt; post-therapy; IP, intensive phase; BMI, body mass index; LBM, lean body mass; FFM, fat-free mass; BIA, electric bioimpedance; BMD, bone mineral density; BGPPY, protein PPY; BGL, blood glucose level; BG; blood glucose; RDI, recommended dietary intake; GBTLI-ALL, Brazilian group for treatment of acute lymphocytic leukaemia in infancy; DFCI, Dana–Faber Cancer Institute; JCLSG, Japanese Children’s Leukaemia Study Group; CRP, *C*-reactive protein; I1, induction; I2, induction; I2A, induction; I2, induction 2; M6, maintenance 6 cycles; GLATHEM, Grupo Latinoamericano de Tratamiento de Hemopatías Malignas; HOPDA, Haematology/Oncology Pediatric Department of the University of Athens. DCLSG: protocol of de Dutch Childhood leukaemia Study Group; EORTC, European Organization for Research and Treatment of Cancer trial; MRCUK, Medical Research Council UK trials.

## 3. Results

Overall, 1180 articles were extracted. We excluded 1115 articles after eliminating duplicates (*n* = 201) and reviewing titles and abstracts. After full test analysis, 65 articles remained, of which 47 were eliminated because they did not fit the research topics. Finally, this systematic review was based on 18 publications (Figure 1).

### 3.1. Study Characteristics

The most important characteristics of the studies included are detailed in Table 3.

Of the 18 articles, trials with different designs were identified, although all were longitudinal studies except for 1. Additionally, all the articles included were clinical trials not randomised. We also found heterogeneous sample sizes within the included studies, with patient groups ranging from 13 to >700. The main similarity between these studies was the age of the patients recruited (1–18 years). The total sample consisted of 1692 participants.

### 3.2. Influence of ALL Treatment in Anthropometric Parameters and Metabolic Biomarkers

Tamminga et al. (1992) [22], Halton et al. (1998) [23], and Sgarbieri et al. (2006) [24] reported a statistically significant delay in growth in all patients with ALL during treatment, especially in those undergoing CI or in patients with HR. This delay was more notable during the first year of treatment [22,23,24], even within the first 6 months, as Tamminga et al. observed [22], being reduced thereafter. In this sense, Halton et al. [23] also observed a further decline in height during the second year, only in children who received CI. After completing the treatment, growth experienced catch-up in both groups [22,23]. No differences were observed in head circumference between the groups [22].

Regarding patients’ body mass index (BMI), Higashiyama et al. (2014) [25] described good basal nutritional status based on BMI in around the 60% of the sample decreasing during the different stages of treatment, although only significant differences were observed in HR patients during the sanctuary and reconsolidation phases. González et al. (2004) [26] analysed anthropometric parameters in a longitudinal study during different phases of treatment, showing that almost 40% of the patients gained weight after the intensive phase and at the end of treatment (especially in boys). No significant differences were observed in the skinfold measurements. Similar observations were reported by Halton et al. (1998) [23] and Tamminga et al. (1992) [22], showing a trend towards obesity, although in this last study, enteral feeding was required for a period of time in some patients. Morrison et al. (2011) [27], in contrast, reported that most of their cohort had a normal BMI at diagnosis, which decreased after intensive therapy (first 6 months) and recovered progressively thereafter.

In addition, Higashiyama et al. (2014) [25] found normal serum albumin levels in their cohort at diagnosis that declined significantly during treatment. Halton et al. (1998) [23] and Gokhale et al. (2007) [28] also reported the same trend. Other metabolic biomarkers, such as cholesterol, retinol, and haemoglobin levels, were also analysed by Gokhale et al. (2007) [28] at maintenance and post-treatment, but did not show significant differences between the groups, although retinol levels were globally lower in patients with ALL.

### 3.3. Influence of ALL Treatment in Digestive Health, Biochemical Parameters and Dietary Intake

Halton et. al. (1998) [23] studied different indicators of digestive health, such as liver enzymes, faecal mass, and absorption of d-xylose or lactose, but did not find data suggestive of generalised malabsorption in these patients.

Regarding the dietary intake of children undergoing ALL, Delbecque et al. (1997) [29] observed reduced energy intake in the patient group at the beginning of treatment (day 1) accompanied by a reduction in carbohydrate intake. These aspects disappeared on days 36 and 71. The respiratory quotient (RQ) was also significantly lower at diagnosis, becoming similar to the controls, when energy intake improved (day 71). However, they did not report significant differences in basal energy expenditure between the ALL and control groups. Bond et al. (1992) [30] reported similar results in their cohort. Moreover, Ladas et al. (2019) [31] also evaluated the dietary intake in two groups of children with ALL at different stages of treatment, showing that dietary patterns are not uniform in different age and sex groups. They observed, in contrast with Delbecque et al. (1997) [29], that the caloric intake decreased during therapy for most age/gender groups, and despite this, the majority of the patients were above the dietary reference intake (DRI) at each timepoint, while only 25% had a caloric intake below these recommendations. Furthermore, 50% exceeded the daily fat intake recommendation, and a small percentage (0–11%) exceeded the carbohydrate intake. Morrison et al. (2011) [27] observed that the fat body mass (FBM) and lean body mass (LBM)m which were normal at diagnosis, were later modified. There was a gain in FBM during the first 6 months of treatment at the expense of a loss in LBM. Furthermore, they concluded that there was a significant correlation between serum creatinine levels and LBM. Sala et al. (2005) [32] observed the same tendency, and also observed that this relationship was not detected between serum creatinine levels and BMI.

When different nutritional biomarkers were analysed, Sgarbieri et al. (2006) [24] reported higher levels of copper in children with ALL, decreasing these levels significantly during antineoplastic treatment. However, zinc levels did not change with treatment despite being globally lower in children with ALL than in healthy children.

Andria et. al. (2020) [33] also reported alterations in glucose metabolism in patients with ALL during treatment, which were more frequent in malnourished patients. Most treatments, such as steroids and their combinations, can modify glucose metabolism depending on individual sensitivity [34]. In accordance with Santoso et al. (2010) [35], these patients undergoing treatment are also at risk of developing hypocalcaemia and/or vitamin D insufficiency (25(OH)D3 (20–30 ng/mL) or deficiency (25(OH)D3 < 20 ng/mL). None of the children presented bone fractures during treatment, although some were diagnosed with osteopenia and osteoporosis.

Roy et al. (2017) [36] described deficient levels of folate at the maintenance phase that seemed not to be associated with dietary patten and had a higher incidence in females. This deficiency was associated with the reduction of the dosage treatment.

Pais et al. (1990) [37] reported lower plasma levels of vitamin B6 in patients with ALL. Plasma levels of plasma pyridoxal 5′-phosphate (PLP) have been positively correlated with dietary intake of vitamin B6 and were not observed to have a significant relationship with lactate dehydrogenase (LDH) levels or bone marrow cellularity.

Regarding the hormones that regulate hunger and satiety, De Carvalho et al. (2020) [38] also reported an increase in energy intake and macronutrients during the induction phase, accompanied by a significant increase in blood ghrelin levels. There were no significant changes in the levels of insulin, cortisol, or leptin. The same trend in regard to ghrelin was observed by Moschovi et al. (2008) [39], as the levels of this hormone were lower at diagnosis compared with the end of maintenance; however, they were still low in comparison with those of the control group. They also saw an increase of peptide YY(PYY) during the first stages of treatment and a decrease to pretreatment levels afterwards

### 3.4. Analysis of Bias in the Systematic Review Studies

Of the eighteen studies selected, eight presented a moderate risk of bias and ten had a serious risk. Depending on the domain, the risk of bias due to confounding variables was moderate or serious in almost all articles. This was because many of these studies did not consider the family socioeconomic level of these children, their daily diet before treatment, or their level of physical activity as factors that could have affected their subsequent nutritional status.

The risk of bias due to participant selection remained moderate in some studies [22,23,24,25,32,35,37,38] because of the high age range of the participants.

Some articles [27,28,29,30,31,33,34,36,39] presented an overall high risk of bias owing to a lack of information about the treatment plan, dosage, or duration.

Furthermore, some studies had a high risk of confounding bias, because the sociodemographic characteristics of the participants were not statistically considered.

The risk of bias of the articles included in this systematic review is summarised in Figure 2 and Figure 3.

## 4. Discussion

In our systematic review, we selected 18 articles, observing that nutritional status is influenced by the treatment, particularly in the first year.

Basal nutritional status and its modifications since diagnosis play an important role in cancer evolution, treatment response, and global prognosis. The illness itself has an impact on the nutritional status, as cancer cells affect three metabolic hallmarks such as increased glycolysis, glutaminolysis, and lipogenic pathways [7], as well as antineoplastic treatments, and the characteristics of each child (social and environmental).

Tools are needed to carry out adequate nutritional assessments, particularly in high-risk patients. We must consider that BMI is not a good predictor of nutritional status because of its inaccuracy when evaluating body composition, as it is well known that BMI does not adequately assess the FBM and LBM [40]. Therefore, it should be considered in combination with other measures, such as anthropometric measurements and blood biomarkers.

Different antineoplastic treatments have been shown to influence growth rates [22,23,24] and the quality and quantity of dietary intake, particularly macronutrient distribution. In a study carried out in 2016 by Kumar et al. [41], the authors observed a deceleration in height throughout therapy, being highest in the first 6 months of therapy, coinciding with Tamminga et al. [22] observations, and catching up thereafter. These results are similar to the observations in this review. They attributed those outcomes to the CI and chemotherapy, as it seems that both therapies had an effect on growth hormone (GH) regulation with GH deficiency and epiphyseal growth plates, respectively. In this respect, the elimination of CI seemed to result in a significant reduction in complications such as growth, delayed pubertal development, endocrinopathies, and neurological impairment [42].

In relation to weight gain in these children, it may also be influenced by corticosteroid treatment [43,44], especially during the first 30 days. Sgarbieri et al. (2006) [19] pointed out that this could be related to an increase in energy intake (conditioned by an excessive desire to eat) and water retention. Halton et al. (1998) [23] explained that an increase in weight may reflect an increase in fat mass in response to GH reducing physical activity during treatment. This decrease in exercise appears to have been due to musculoskeletal morbidity, which caused a reduction in energy expenditure. Regarding the performance of exercise during treatment, the current evidence confirms that it has a positive impact on skeletal bone, neuromuscular, musculoskeletal, cardiovascular, and cardiopulmonary systems, as well as on metabolism alterations and body balance disorders, and even alleviates fatigue linked to medication [44]. For this reason, physical activity is currently recommended from the time of diagnosis.

Regarding serum albumin as a nutritional status biomarker, some authors agree [23,25,28] that albumin levels decrease during ALL treatment. Halton et al. (1998) [23] stated a possible mechanism to explain this fact: a reduction in protein intake as well as a basal hypermetabolic state, which results in the depletion of visceral proteins. Also, Roy et al. (2018) [35] detected folic acid deficiencies in these patients during treatment, which were thought to be related to an increased risk of cytopenia and hypoalbuminemia.

Altered levels of serum Cu and Zn in children with ALL were observed by Sgarbiery et al. (2006) [24], and may have been related to the inflammatory conditions of infectious diseases. Its determination in blood may be a useful tool for detecting the presence of malignancies, although it does not seem to be a sensitive prognostic biomarker.

Andria et al. (2020) [33] and Aisyi et al. (2019) [34] reported that steroids and L-ASNase may cause hyperglycaemia, especially when L-ASNase and dexamethasone are co-administered, which could be related to insulin resistance and metabolic health programming risks [16]. Furthermore, Andria et al. (2020) [33] concluded that abnormal glycaemia during the intensive phase of chemotherapy did not affect remission status.

Regarding hypocalcaemia and hypovitaminosis D observed by Santoso et al. (2010) [35], they may have been caused by continued exposure to corticosteroids, leading to inhibition of the synthesis of 1,25(OH)2D3 and a deficit in calcium absorption.

De Carvalho et al. (2020) [38] found an increase in blood ghrelin levels, a hormone related to food intake regulation, which may be linked to chronic exposure to low-calorie diets, although not in this case, as children showed adequate energy intake. However, ghrelin levels are thought to decrease at the time of ALL diagnosis when abnormal inflammatory indices are present.

PLP reflects the total body vitamin B6 levels. Pais et al. (1990) [37] observed that the levels were significantly low in patients with ALL. The authors believe that PLP levels are the result of the disease, instead of the cause, which can be explained by decreased vitamin B6 intake during illness, decreased hepatic conversion of pyridoxine to PLP, or larger requirements for vitamin B6 by neoplastic cells. Moreover, the authors considered that this deficiency could aggravate treatment toxicity.

All things considered, it seems that the metabolism of the disease itself, as well as the ALL treatments, affect the children’s intake and the regulation of hormonal and metabolic pathways, as it was evidenced by altered micro and macronutrients in this review, resulting in impaired growth rates. According to our data, the highest-risk phase was the first year of treatment, probably due to the therapeutic scheme and its dosage intensity.

Consequently, nutritional counselling became a key intervention during the treatment. Good nutritional counselling for these paediatric patients is not only important for the current maintenance of an adequate nutritional status, but is also important for adult health. Some studies have linked survivors of some types of paediatric cancer, including ALL, to a higher risk of metabolic syndrome in adulthood. These patients have higher risk factors for cardiovascular diseases, such as increased visceral fat, insulin resistance, dyslipidaemia, and hypertension [45,46]. It has been shown that healthy prepubertal patients with obesity and elevated levels of cardiovascular biomarkers are more likely to develop metabolic syndrome later in life [15,16].

Nutritional status at the start of treatment has been described as a prognostic factor of survival in patients with ALL. According to this, Antillon et al. [47,48] determined that severe malnutrition at diagnosis was associated with higher rates of both abandonment of treatment and relapse of the disease, and also that the nutritional status at 6 months of treatment was a good prognostic factor for survival. Moreover, Den Hoed et al. [49] linked low weight at the time of diagnosis to lower survival rates and excessive weight gain during treatment to worse disease prognosis. In line with these data, in a systematic review, it was observed that ALL survivors showed greater tendency towards obesity than healthy children at the same age and gender, and this occurred regardless of treatment, gender, or age at the diagnostic [50]. This fact has been linked with lower survival rates and greater toxicities related to treatment [51]. Owing to the greater increase in weight occurring when the treatment started, Walters et al. (2021) [52] carried out the first nutritional intervention study in ALL paediatric patients from the beginning of the induction phase. It consisted of recommending a low-glycaemic diet for 6 months. They concluded that patients showed adherence to the consumption of highly nutritious foods that are linked to the prevention of obesity, so the application of these interventions is feasible for preventing weight gain during treatment. Recently, Guzman et al. (2024) [53] published a systematic review that studied the effect of numerous nutrition interventions on improving nutritional status and body composition in ALL children under treatment. Although none of the interventions seemed have been consistently positive in nutritional status, they observed some good results in terms of hospitalisation days, presence of oedema, neuropathy, recovery time, haemoglobin, and gastrointestinal discomfort with the use of glutamine, honey, black seed oil, and probiotics.

The socioeconomic and educational levels of parents and their lifestyles are different modifiable factors that could influence basal nutritional status throughout treatment. Owing to the importance of basal nutritional status, dietary intake, and healthy lifestyle in the evolution, response to treatment, and prognosis of children with ALL, nutritional interventions should be one of the pillars of multidisciplinary treatment. For these reasons, it is of great importance at the time of diagnosis and during the different phases of treatment, especially in the first year, to perform a personalised nutritional intervention by specialised health personnel.

The main limitation of this review is its heterogeneity, including the wide age interval of the sample. Moreover, the sample size ranged from less than 20 patients to almost 800. On the other hand, the protocols used were also heterogeneous, as many different treatment regimens were used in the 18 selected publications, and some of them were even outdated. In addition, we must highlight the lack of valuable information to analyse the results of some articles, as evidenced in the risk of bias.

## 5. Conclusions

The basal nutritional status at diagnosis and during antineoplastic treatment is a prognostic factor for the global survival of patients with ALL. Some treatments for ALL, such as corticosteroids, can modify healthy lifestyles or global intake because of their implications for hunger regulation, affecting the nutritional status of paediatric patients.

A delay in growth, especially in the first year of treatment, was observed. In addition, alterations in micro and macronutrients were observed, such as decreased albumin and copper levels, increased ghrelin levels, and deficiencies in calcium, vitamin D, and folic acid.

Therefore, nutritional assessment and specific nutritional interventions need to be performed in these patients to decrease their nutritional risk and improve their prognosis and long-term health. This could include actions to prevent obesity, undernourishment, or nutrient deficiencies. In some cases, these measures may involve the use of nutritional supplements, as well as enteral or parenteral feeding.

It is necessary to follow a clinical practice guideline to standardise the nutritional assessment and interventions, based on the patient’s nutritional status and characteristics, and the type of treatment, i.e., “personalised nutritional intervention”.

Homogeneous and well-designed interventional longitudinal studies must be performed to determine the type of nutritional intervention that is optimal to guide ALL prevention, treatment, and prognostic strategies.

## Figures and Tables

**Figure 1 nutrients-16-04119-f001:**
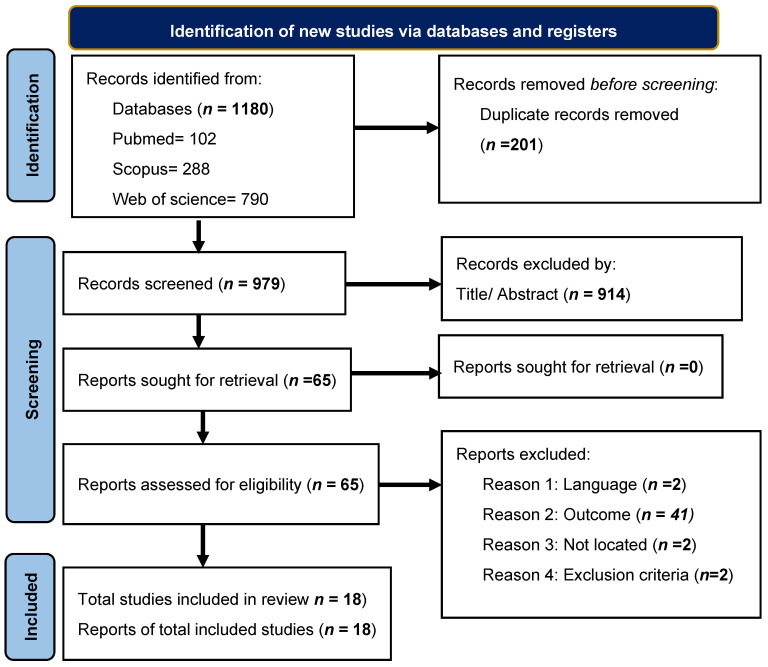
Flowchart of article extraction by PRISMA method.

**Figure 2 nutrients-16-04119-f002:**
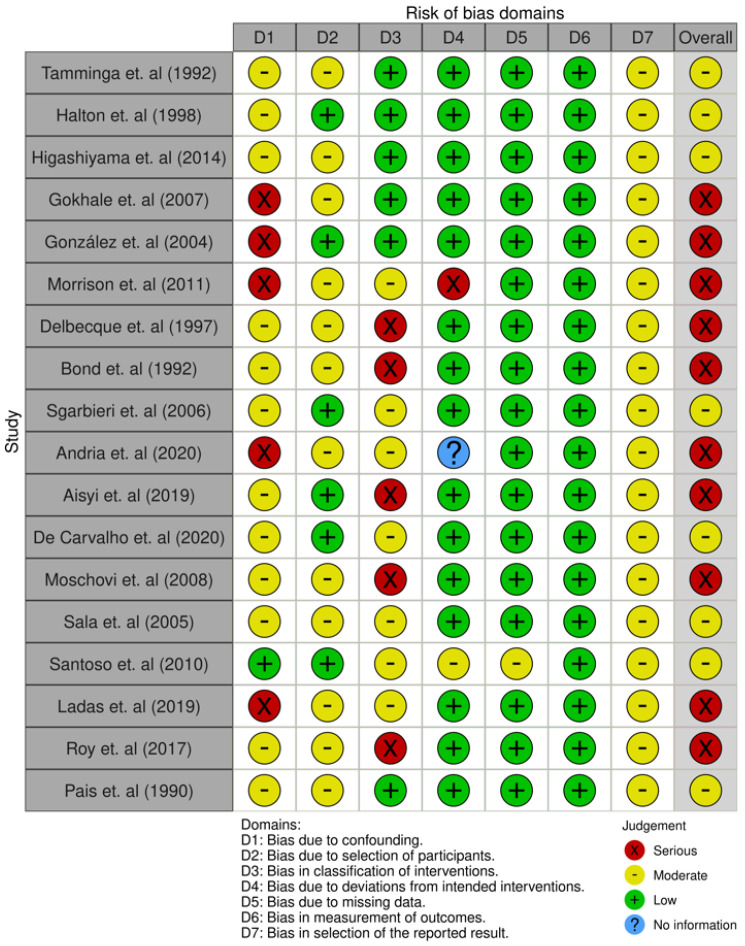
Result of the risk of bias assessment of the study articles [22,23,24,25,26,27,28,29,30,31,32,33,34,35,36,37,38,39].

**Figure 3 nutrients-16-04119-f003:**
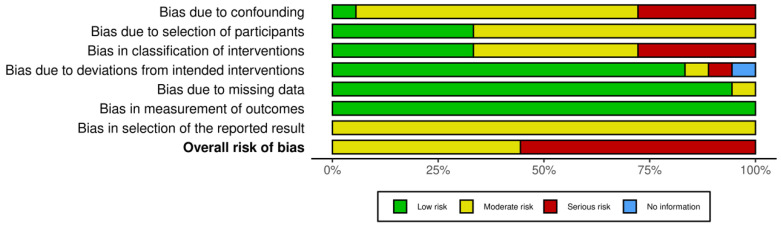
Summary of risk of bias assessment of the study articles.

**Table 1 nutrients-16-04119-t001:** Eligibility criteria based on the PICO elements.

Inclusion Criteria	
Population	Paediatric patients with ALL diagnosis under 18 years.
Intervention	Different types and protocols of treatment for ALL
Comparison	Groups of children and adolescents under different protocols and/or phases of treatment for ALL
Outcome	Nutritional status based on the analysis of anthropometric and biochemical and metabolic biomarkers

Abbreviations: ALL, acute lymphoblastic leukaemia.

**Table 2 nutrients-16-04119-t002:** Search formulas of principal databases.

Database	Search Formula
PubMed	“Leukemia”[Mesh] OR “Acute Lymphoblastic Leukemia”[Mesh] AND (“Child”[Mesh] OR “Infant”[Mesh] OR “Adolescent”[Mesh] OR “Child, Preschool”[Mesh]) AND (“Nutritional Status”[Mesh] OR “Nutrition Assessment”[Mesh]))
Scopus	TITLE-ABS-KEY (“acute lymphoblastic leukemia” OR “leukemia”) AND (“children *” AND NOT “adult *”) AND (“nutritional status *” OR “nutritional assessment”)
Web of Science	Leukemia OR Acute lymphoblastic leukemia AND (child * OR adolescent * OR infant *) AND (“nutritional status” OR “nutritional assessment”)

* Capture variations of the word roots.

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
