# Peer review of "Impact of Acute Lymphoblastic Leukaemia Treatment on the Nutritional Status of Paediatric Patients: A Systematic Review"

_nutrients, 2024, doi:10.3390/nu16234119_

Round 1

Reviewer 1 Report (New Reviewer)

Comments and Suggestions for Authors

The Authors present a paper: " Impact of acute lymphoblastic leukemia treatment on the nutritional status of pediatric patients. A Systematic Review" of low interest for the mentioned too many biases by Authors themselves. The paper is well written and documented but the references are not updated (many of them are quite old) and related to dome treatment protocols that changed over the time (e.g. radiotherapy now excluded in the majority of ALL Children). I am not convinced that nutritional status could predict problems in adulthood as for example cranial irradiation. The Authors say that physical exercise not performed since the onset of the disease could worsen the nutritional status but now all the ALL children benefit of sport activity since the diagnosis. The paper is a review not easy to follow as well as the reported tables. I wonder what is the rea take home message from this paper?  "Nutritional assessments and specific nutritional interventions need to be performed in these patients to decrease their nutritional risk": In which way? they remain too vague and they relaunch to future well-designed studies. I wonder again: what is the value of this review?

Author Response

Dear Reviewer, thank you very much for your comments regarding our manuscript. We truly appreciate your recommendations as we consider they improve the quality of our work. We have remarked all the changes so that you can see them easily.

The Authors present a paper: "Impact of acute lymphoblastic leukemia treatment on the nutritional status of paediatric patients. A Systematic Review" of low interest for the mentioned too many biases by Authors themselves. The paper is well written and documented but the references are not updated (many of them are quite old) and related to some treatment protocols that changed over the time (e.g. radiotherapy now excluded in the majority of ALL Children). I am not convinced that nutritional status could predict problems in adulthood as for example cranial irradiation. The Authors say that physical exercise not performed since the onset of the disease could worsen the nutritional status but now all the ALL children benefit of sport activity since the diagnosis. The paper is a review not easy to follow as well as the reported tables. I wonder what is the real take home message from this paper? "Nutritional assessments and specific nutritional interventions need to be performed in these patients to decrease their nutritional risk": In which way? They remain too vague and they relaunch to future well-designed studies. I wonder again: what is the value of this review?

Dear Reviewer, thank you very much for your enriching comments, which we are sure have helped us to improve our article.

  1. In relation to the age of the articles, as you well know, the objective of this systematic review is to evaluate the effect that the different ALL treatments have on the nutritional status of children. Our review includes all clinical trials until December 31th 2023 that meet the established inclusion and exclusion criteria. Therefore, some of the treatments of the 18 selected articles are no longer included in current protocols, but we believe that they provide important information for the advancement to propose new future protocols.
  2. Regarding the question about how the nutritional status can predict problems in adulthood, we believe that the role of healthy lifestyles is well documented in the scientific literature. Healthy lifestyles, which are now associated with the main causes of morbidity and mortality, and nutritional status in paediatric age are related to metabolic and cardiovascular health in the short, medium and long term. A major line of research at present is how the critical periods of growth and development (1,000 first days of life, infancy and adolescence) are windows of opportunity for adult health prevention. Adequate nutrition and development, as well as microbiota, are associated with multiple pathologies.

In order to clarify it, we added some changes on the text. Please, find them (pag.2, section 1 Introduction).

“Studies have evidenced that ALL therapies in childhood are linked to a greater risk of chronic metabolic diseases later in life, probably increased, among other reasons, as consequence of the weight gain and obesity. Also, it was observed that these conditions are present fundamentally in patients with overweight/obesity at the diagnosis and which remain with a high BMI z-score during the treatment and until the beginning of survival [13]”.

“Specifically, corticosteroids and CI used in protocols against ALL, are well known to be related to the regulation of energy intake and impair signalling reception from hormones that regulate hunger and appetite. This, results in several effects on body composition, leading to short- and long-term impact on nutritional status of survivors [11]. Also, as consequence of the treatment, the motor function of ALL patients becomes impaired, extending this condition to adolescence and adult age [12], what difficult the physical activity practice”.

  1. Regarding the role of physical activity, we fully agree with the reviewer's comment. Today, many children with ALL are advised to practice sports from diagnosis. Therefore, the results of the Halton study [19], included in our review support this recommendation. We added the following sentence in the discussion to reflect the reviewer's commentary. Please, find them (Pag.23, Section 4 Discussion)

“Concerning the performance of exercise during the treatment, the current evidence confirmed that it has positively impact on skeletal bone, neuromuscular, musculoskeletal, cardiovascular and cardiopulmonary systems, as well as the on metabolism alterations, body balance disorders and even alleviating the fatigue linked to medication [45]. For this reason, physical activity is currently recommended from the time of diagnosis.”

  1. With the objective of offer you a response to the take-home message, we have added the following paragraph. Please, find it (Pag.25, Section Discussion 4).

“For these reasons, it is of great importance at the time of diagnosis and during the different phases of treatment, especially in the first year, the performance of a personalized nutritional intervention by specialized health personnel.”.

  1. 5. Dear reviewer, in order to answer your cuestion about the value of our revision, we would like to stand out, that at the present time, malnutrition disease-related is a major health challenge in developed countries. We believe that this review emphasizes the importance of identifying the risks of undernutrition, preventing it, diagnosing it and and establish intervention strategies within ALL treatment protocols. It highlights the main biomarkers affected and the possible causes to guide our strategies. causes to guide our strategies. To clarify the value of this work, we have added the following paragraph. Please, find it (Pag. 3, Section 1 Introducction)

“Besides, undernourishment in the paediatric age in industrialised countries is now associated with the disease. For this reason, is essential to highlight the necessity an adequate adherence of clinic practice guidelines for nutrition and physical activity in children with ALL.

  1. In response to the reviewer's suggestion about improving the research design and description of methods. We have:

- Modified the PICOS table.

- Reformulated the research questions.

- Modified the flow chart with the new results of the application of the research questions.

- Modified Table 3.

Please, find these changes (pag.3, section 2 Material and Methods: Table 1 and Table 2; pag.4, Section 3 Results: Fig. 1; pag. 8, Results: Table 3)

As you can observe on the text, the final result, in terms of the 18 articles selected, the results and conclusions, has not been modified. We hope that these changes respond to the reviewer's suggestions.

We hope all your questions and comments you required us had been solved satisfactorily. We truly believe the modifications made were necessary and they totally improve the quality of this revision. In any case, we remain at your full disposal for any further suggestion.

Reviewer 2 Report (New Reviewer)

Comments and Suggestions for Authors

ALL is the most prevalent type of cancer and the leading cause of cancer-related deaths in children, and considering the targeted population group, it is important to establish and achieve the most suitable nutritional status. Furthermore, this pathology could alter the palatability and perception of foods, increasing the risk for malnutrition and consequently to a normal growth. Therefore, the present review could highlight the most significant aspects of a better assessment of the nutritional status in children with ALL. The manuscript is a comprehensive review of the literature database focused on this topic, with a well-documented and presented methodology, including a well designed PRISMA chart. For this reason, the results are clearly described and well detailed, and the conclusions are in accordance with the presented data. I congratulate the authors for a detailed review that could improve our current knowledge.

Author Response

Dear reviewer. Thank you very much for your kind comments about our work. We are truly grateful for them.

Reviewer 3 Report (New Reviewer)

Comments and Suggestions for Authors

The paper is quite well writtena and presented, although with relatively little novely and originality.

In introduction there is no presentation of the causes of poor nutritional status and metabolic disturbances in ALL.

Are there any studies of nutritional interventions?

Could the authors relate to stdies of individuals post ALL in treatment in terms of long-term metabolic and e.g. cardiovascular morbidity?

Also in discussion and conclusions that could be useful to suggest how the nutritional status could be targeted /improved and what are the future directions.

Are there any international recommendations?

Author Response

Dear Reviewer, thank you very much for your comments regarding our manuscript. We truly appreciate your recommendations as we consider they improve the quality of our work. We have remarked all the changes so that you can see them easily.

The paper is quite well written and presented, although with relatively little novelty and originality.

Dear author, thank you very much for this appreciation. We absolutely agree with you that some articles included are outdated, however we still consider we had to include them as they met our inclusion criteria. This point it is already indicated in the limitations section. In addition, we have tried to include the most recent articles for the introduction and discussion to offer the newest evidence.

  1. In introduction there is no presentation of the causes of poor nutritional status and metabolic disturbances in ALL.

Dear reviewer. Thank you very much for your valuable comment. According your suggestion, we have introduced some changes on the text. Please, find them on the text (Pag. 2, section 1 Introduction)

“The disease itself has a negative effect on the nutritional status as cancer cells involve alterations on some metabolic pathways such as glycolysis, glutamonolysis and the lipogenesis [7]”.

“Specifically, corticosteroids and CI used in protocols against ALL, are well known to be related to the regulation of energy intake and impairs signalling reception from hormones that regulate hunger and appetite. This, results in several effects on body composition, leading to short- and long-term impact on nutritional status of survivors [11]. Also, as consequence of the treatment, the motor function of ALL patients becomes impaired, extending this condition to adolescence and adult age [12], what difficult the physical activity practice”.

  1. Are there any studies of nutritional interventions?

Dear reviewer, thank you very much for this question. In our systematic review we have set the use of nutritional interventions as an exclusion criteria.  However, in response to your suggestion, we have made some changes on the text to answer these queries, as well as your next question, the number 4. Please, find them (Pag. 24-25. section 4 Discussion)

  1. 4. Could the authors relate to studies of individuals post ALL in treatment in terms of long-term metabolic and e.g. cardiovascular morbidity?

“In line with these data, in a systematic review, was observed that ALL survivors show greater tendency to obesity than healthy children at the same age and gender, and this occurs regardless of treatment, gender or age at the diagnostic [51]. This fact has been linked with lower survival rates and greater toxicities related to treatment [52]. Due to the greater increased of weight happens when the treatment starts, Walters et al. (2021) [53] carried out the first nutritional intervention study in ALL paediatric patients from the beginning of the induction phase. It consisted in recommending a low glycemic diet for 6 months. They concluded that patients showed adherence to the consumption of highly nutritious food which are linked to the prevention of obesity, so that, the application of these interventions are feasible to prevent weight gain during treatment. Recently, Guzman et al. (2024) [54] published a systematic review which studied the effect of numerous nutrition intervention to improve nutritional status and body composition in ALL children under treatment. Although none of the interventions seems have been consistently positive in nutritional status, they observed some good results in terms of hospitalization days, presence of edema, neuropathy, recovery time, hemoglobin, and gastrointestinal discomfort with the use of glutamine, honey, black seed oil and probiotics.

  1. Also, in discussion and conclusions that could be useful to suggest how the nutritional status could be targeted /improved and what are the future directions.

Dear reviewer. Thank you very much for your valuable comment. According your question, we have made some changes on the text. Please, find them (Pag. 25, section 5 Conclusion)

Therefore, nutritional assessment and specific nutritional interventions need to be performed in these patients to decrease their nutritional risk and improve the prognosis and long-time health. This could include actions to prevent both the obesity, undernourishment or nutrient deficiencies. In some cases, these measures may imply the use of nutritional supplements, as well as enteral or parenteral feeding.

It is necessary a clinical practice guideline to standardize the nutritional assessment and interventions, based on the patient nutritional status and characteristics, and the type of treatment, “personalized nutritional intervention”.

  1. Are there any international recommendations?

Dear reviewer, thank you very much for this question. We introduced a brief comment on the text (Pag. 2, section 1 Introduction)

“For this reason, is essential to highlight the necessity an adequate adherence of clinic practice guidelines for nutrition and physical activity in children with ALL”.

  1. In response to the reviewer's suggestion about improving the research design and description of methods. We have:

- Modified the PICOS table.

- Reformulated the research questions.

- Modified the flow chart with the new results of the application of the research questions.

- Modified Table 3.

Please, find these changes (pag.3, section 2 Material and Methods: Table 1 and Table 2; pag.4, Section 3 Results: Fig. 1; pag. 8, Results: Table 3)

As you can observe on the text, the final result, in terms of the 18 articles selected, the results and conclusions, has not been modified. We hope that these changes respond to the reviewer's suggestions.

We hope all your questions and comments you required us had been solved satisfactorily. We truly believe the modifications made were necessary and they totally improve the quality of this revision. In any case, we remain at your full disposal for any further suggestion.

Round 2

Reviewer 1 Report (New Reviewer)

Comments and Suggestions for Authors

The Authors re-written the manuscript in a more complete and justified way following the reviewers suggestions.

Reviewer 3 Report (New Reviewer)

Comments and Suggestions for Authors

The paper is now improved and satisfactory for publication.

This manuscript is a resubmission of an earlier submission. The following is a list of the peer review reports and author responses from that submission.

Round 1

Reviewer 1 Report

Comments and Suggestions for Authors

Dear Authors

Congratulations for your work and efforts.

I read your submission carefully. It presents critical weaknesses which must be fixed.

My comments:
1) The PICO is not correct. You do not describe clearly the treatment for ALL. You know that it is heterogeneous now as well as across time. So it must be more clear. Also, you wrote the comparison of the "nutritional status", and it was not correct. The comparator needs to be what the treatment will compare to. Another weakness is the changes in nutritional status that you want to analyse.

2)The inclusion and exclusion criteria did not reflect the PICO. Also, for this type of systematic review, you must integrate the study designs as an inclusion criterion.

3) The search syntaxes in the different databases are not equivalent. It is a huge weakness and bias.

3)You wrote in the 2.4-Extraction data the information "type of intervention", which is not the respective table.

4) The results must be presented by type of treatment and not by outcome,

5) Discussion: there are no study limitations, and the bias was not considered in the discussion.

6) Conclusions: the primary outcomes analysed must be more precise.

In conclusion, you need to make a significant revision.

Kind regards

Reviewer 2 Report

Comments and Suggestions for Authors

Nutritional status in children with cancaer is an important prognostic factor. I read with the interest a systematic review on the impact of antileukemic treatment on the nutritional status in children, based on 20 published articles.

The main flaw are heterogeneous populations / subsets of patients with ALL, different risk group (which the author state a an important influence), different risk-adapted treatments and different phases of treatment. The impact of cranial radiotherapy is highlighted, but it is usually not in the "early stages" as mentioned but at the beginning of the maintenace phase. There is a difference between prophylactic and therapeutic cranial radiotherapy too.  Different measurements have been included. Furthermore, different biomarkers including vitamins and hormones, were measured. Some articles lack the information about patients' demographic characteristics and some about the treatment, which impact on the nutritional status is the main question. Basal nutritional status is missing, as well as the explanation of the catabolyc state of cancer itself. What about the nutritional intervention in published studies?

Introduction section should be focused, instead the explanations about helthy lifestyle and nutritional habits in pediatric age.

Where children < 18 years included or from 1 - 18 as stated later in the text?

Comments on the Quality of English Language

Required editing.

Usage of more appropriate terms is required, eg. highest concentration of ALL - the peak incidence; impact on treatment scheme - risk stratification; what means "the intensity of drugs"?